# Early Motor Trajectories Predict Motor but not Cognitive Function in Preterm- and Term-Born Adults without Pre-existing Neurological Conditions

**DOI:** 10.3390/ijerph17093258

**Published:** 2020-05-07

**Authors:** Nicole Baumann, James Tresilian, Peter Bartmann, Dieter Wolke

**Affiliations:** 1Department of Psychology, University of Warwick, Coventry CV4 7AL, UK; N.Baumann.1@warwick.ac.uk (N.B.); J.R.Tresilian@warwick.ac.uk (J.T.); 2Department of Neonatology, University Hospital Bonn, 53113 Bonn, Germany; Peter.Bartmann@ukbonn.de; 3Warwick Medical School, University of Warwick, Coventry CV4 7HL, UK

**Keywords:** longitudinal studies, motor development, preterm birth, IQ, adulthood, cognitive function

## Abstract

Very preterm (VP; <32 weeks gestation) and/or very low birth weight (VLBW; <1500 g) birth has been associated with an increased risk of adverse motor and cognitive outcomes that may persist into adulthood. The aim of this study was to determine whether motor development in the first five years of life is associated with motor and cognitive outcomes in adulthood. A prospective observational study in Germany followed 260 VP/VLBW and 229 term-born individuals from birth into adulthood. Early motor trajectories (i.e., high and low degree of motor difficulties) were determined from neurological examinations from birth to 56 months. Adult motor and cognitive outcomes were determined from information from multiple instruments and IQ tests, respectively. Associations of VP/VLBW birth and early motor difficulties on adult outcomes were assessed using regression analyses. VP/VLBW individuals had an increased risk for early motor difficulties (Relative Risk: 11.77, 95% confidence interval (CI): 4.28, 32.35). Early motor difficulties were associated with poorer motor competence in adulthood (*β* = 0.22, *p* < 0.001), independent of VP/VLBW birth. Adult IQ was predicted by VP/VLBW (*β* = −0.12, *p <* 0.05) and child IQ (*β* = 0.51, *p <* 0.001), while early motor difficulties ceased to be associated with adult IQ once participants with a neurological impairment were excluded (*β* = 0.02, *p* > 0.05). Motor problems in childhood were homotypically associated with poorer motor competence in adulthood. Similarly, early cognitive problems were homotypically associated with adult cognitive outcomes. Thus, both motor and cognitive function should be assessed in routine follow-up during childhood.

## 1. Introduction

Despite advances in antenatal and neonatal care over recent decades, very preterm birth (VP; gestational age <32 weeks) and/or very low birth weight (VLBW; <1500 g) remain major risk factors for neurodevelopmental sequelae, including motor and cognitive impairments [1,2]. Both motor and cognitive impairments can have a profound impact on learning, academic achievement, health, and wellbeing across the lifespan for VP/VLBW individuals [3,4], and may persist throughout childhood and into adulthood [5,6,7,8]. Furthermore, they are the two most frequent sequelae of VP/VLBW [9].

According to Piaget and more recent embodied approaches [10], cognitive development in the first two to three years proceeds through sensorimotor interactions between the child and its physical and social environment. Consistent with these ideas, it has been found that a number of cognitive tasks recruit the same brain regions that are used for motor tasks [11,12]. Numerous cross-sectional studies have reported an association between motor and cognitive function in typically developing [13,14,15,16] and preterm-born children [17,18]. This raises a question of whether associations between the motor and cognitive domain will hold in a longitudinal setting: Does children’s early motor development affect their cognitive abilities as they grow into adults? Or is there a homotypic association where early motor development specifically predicts later motor outcomes but not cognition? Several longitudinal studies have shown that early motor difficulties predict poorer cognitive abilities into school-age, albeit less consistently in preterm than in general or term-born populations [19]. However, long-term studies that investigate associations between early motor development and later cognitive abilities into adulthood are rare [20,21,22,23], particularly in preterm populations.

The current study investigated a sample of VP/VLBW and control individuals from birth to adulthood. The study aims were to investigate: (1) whether VP/VLBW birth is associated with early motor development from birth to 56 months (i.e., early motor trajectories), and (2) whether VP/VLBW birth and early motor trajectories are specifically associated with motor competence in adulthood or with adult cognitive function.

## 2. Methods

### Design and Participants

Data were collected as part of the prospective Bavarian Longitudinal Study (BLS) [24]. The BLS is a geographically defined, whole population sample of neonatal at-risk children born in 1985 and 1986 in Southern Bavaria, Germany who required admission to a children’s hospital within the first 10 days after birth (*n* = 7505). The initial cohort included 682 participants born VP/VLBW. Of these, 411 were alive and eligible for follow-up in adulthood, and 260 (63.3%) participated. Additionally, healthy infants born at term in the same hospitals were recruited as controls (*n* = 916). To be comparable to the VP/VLBW sample, 350 participants of this group who were alive at 6 years were randomly selected as term controls within the stratification variables gender and family socioeconomic status (SES). In adulthood, 308 term-born individuals were eligible for inclusion and 229 (74.4%) participated at 26 years (see Figure 1). The current study used data collected at birth, 5, 20, and 56 months, and 26 years. Participants were assessed for one whole day, including neurological testing in childhood and cognitive testing and detailed interviews at all time points.

All assessors were blind to participants’ characteristics and results of previous assessments. Ethical permissions were granted by the ethics committees of the University of Munich Children’s Hospital and the Bavarian Health Council (Landesärztekammer Bayern). Ethical approval for follow-up in adulthood was granted by the Ethical Board of the University Hospital Bonn (reference # 159/09). Informed written consent was provided initially by parents within 48 h of the infant’s hospital admission and all participants gave fully informed written consent for the assessment in adulthood.

## 3. Measures

### 3.1. Early Motor Trajectories

At birth, and at 5, 20, and 56 months, motor functioning was measured via standard neurological and physical examinations based on Prechtl’s neurological examination method [25]. To compute a motor problem score for each age, items on neurological and motor functioning, including fine and gross motor skills, oculomotor function, muscle tone, and reflexes (for example: eye movement, mouth and tongue movement, quantity of spontaneous motor activity, muscle tone at rest, head control while pulling up to sit, posture during sitting, fluent and stable movements while walking, knee jerk reflex), were combined and categorized into ‘within’ and ‘outside the normal range’ of motor function. Detailed descriptions of items can be found elsewhere [26]. Based on the sum of motor functioning ‘outside the normal range’ a motor problem score was computed, with higher scores indicating more motor problems. Due to the varied number of items across time points (birth: 25 items, 5 months: 63 items, 20 months: 63 items, and 56 months: 125 items), motor problem scores were standardized to a mean of 0 and a standard deviation of 1 (z-scores), allowing comparability across time points and ages.

Using latent class growth analysis (LCGA), a person-centered statistical approach, two groups of children with a similar development of motor functioning from birth to 56 months (i.e., motor trajectories) were identified. LCGA was applied to the sample available in early childhood, independent of birth status (e.g., preterm birth). The details of the statistical procedure are described elsewhere [26]. The two motor trajectories described as either high or low degree of motor difficulties, yielded a binary variable. For the current longitudinal sample 44 (10.5%) and 374 (89.5%) participants had a high and low degree of motor difficulties in early childhood, respectively.

### 3.2. Outcomes in Adulthood

#### 3.2.1. Motor Competence

Indicators of motor function were assessed as part of a life course interview and questionnaires with the participants, including suboptimal functioning for ambulation and dexterity, low self-esteem regarding sports, being poorly coordinated or clumsy, and not being able to ride a bicycle or to swim (see Table A1). Indicators were summed into a composite score. Due to high skewness, the score was dichotomized: 0 (high motor competence) and 1 (low motor competence). Data on adult motor competence were available for 418 participants.

#### 3.2.2. Cognitive Function

General intelligence was assessed with the German short version of the Wechsler Adult Intelligence Scale (WAIS III) [27,28]. Raw scores for six subtests of the WAIS (i.e., vocabulary, similarities, letter-number-sequence, block design, matrix reasoning, and digit symbol coding) were converted into an age-normed full-scale IQ score [29]. Data on adult IQ were available for 354 participants.

### 3.3. Covariates and Potential Confounds

Data on gestational age (GA, weeks), birth weight (grams), and sex (1 = male, 0 = female) were obtained from perinatal records at birth. Children with a birth weight of less than the sex specific 10th percentile for GA were classified as small for GA (SGA) [29]. Family socioeconomic status (SES) at birth was defined and categorized as 1 (low), 2 (middle), or 3 (high) [24].

IQ at 56 months was assessed through a composite of three cognitive tasks (the Columbia Mental Maturity Scale [30,31], Active Vocabulary Test [32], and the Beery–Buktenica Developmental Test of Visual-Motor Integration [33]).

Diagnoses of a range of mild to severe neurological impairments were made by a specially trained developmental pediatrician at 56 months, and included epilepsy, hydrocephalus [24], cerebral palsy (CP; four stages: 1 = more pronounced in fast-paced movement, impairment has a hardly noticeable negative impact (mildest CP); 2 = possible to walk but with noticeable negative impact, additional impairment to the motor function of the hand; 3 = walking is not possible; 4 = no active movement at all (most severe CP)) [34], and blindness, or deafness (not corrected or insufficiently corrected) [35].

### 3.4. Statistical Analyses

All statistical analyses were performed using SPSS Version 22 (IBM SPSS Statistics, IBM Corporation, Armonk, NY, USA) and Stata Version 15.0 (StataCorp LLC, College Station, TX, USA). Differences between VP/VLBW and term-born individuals were tested with a *t*-test for continuous variables and chi-square test for categorical variables. Statistical significance was set at *p* < 0.05.

In addition to the computation of frequencies and group differences, the relative risk (RR) with 95% confidence interval (CI) was calculated to test whether VP/VLBW individuals more often had a trajectory of high degree of motor difficulties in early childhood (reference: low degree of motor difficulties) compared to term-born participants. RR was subsequently adjusted for potential confounds of motor outcomes [9], such as SGA, multiple births, child sex, family SES at birth, and IQ at 56 months.

The weight least square estimation (WLSMV) in Mplus 7 (Muthen and Muthen, Los Angeles, California, USA) was used for structural equation modelling (SEM). Fit of the models was tested using chi-square test of model fit, the root mean squared error of approximation (RMSEA), and the comparative fit index (CFI). Model fits were considered good with RMSEA below 0.06 and CFI above 0.95 [36]. SEM was applied to test direct and indirect effects of early motor difficulties and VP/VLBW on motor and cognitive outcomes in adulthood. The SEM model additionally included IQ at 56 months and was controlled for potential confounds.

To rule out the possibility that the tested longitudinal associations had been influenced by a neurological impairment (*n* = 25; Table 1), a sensitivity analysis was carried out where the SEM model was repeated excluding individuals with a diagnosed neurological impairment in childhood (i.e., epilepsy, hydrocephalus, CP, blindness, or deafness).

## 4. Results

VP/VLBW participants were born with lower GA and birth weight than term-born controls (Table 1). Further, VP/VLBW adults were more often multiples and born SGA. While there were no group differences in regard to sex, VP/VLBW participants were more often born into families of lower and middle SES. At age 56 months, VP/VLBW individuals were more frequently diagnosed with CP and had lower IQ scores. In adulthood, VP/VLBW participants had lower motor competence and lower IQ scores compared to term-born adults (Table 1).

### 4.1. VP/VLBW Birth and Early Motor Difficulties

Compared to term-born individuals, VP/VLBW participants were more likely to have motor difficulties from birth to 56 months (RR: 11.77, 95% CI: 4.28–32.35). This increased risk remained after adjusting for confounds (adjusted RR: 8.32, 95% CI: 2.83–24.44; significant covariates: multiple birth and IQ at 56 months).

After excluding participants with a neurological impairment (VP/VLBW: *n* = 23; term controls: *n* = 2), VP/VLBW participants still had a substantially increased risk for early motor difficulties (RR: 6.96, 95% CI: 2.43–19.92), compared to term-born individuals; even when adjusting for potential confounds (adjusted RR: 4.47, 95% CI: 1.45–13.77; significant covariates: multiple birth and IQ at 56 months).

#### Effects of VP/VLBW Birth and Early Motor Difficulties on Motor and Cognitive Outcomes in Adulthood

SEM results (Model 1) showed that individuals with early motor difficulties were more likely to exhibit lower motor competence in adulthood (*β* = 0.22, *p* < 0.001), while VP/VLBW was not associated with adult motor competence (*β* = 0.13, *p =* 0.151), after accounting for IQ at 56 months, SGA, multiple births, child sex, and SES (Table 2 and Figure 2). Further, child sex (male) negatively affected motor competence at 26 years (*β* = −0.16, *p =* 0.032). VP/VLBW (*β* = 0.35, *p < 0*.001) and multiple births (*β* = −0.19, *p =* 0.002) were associated with early motor difficulties.

Adult IQ was associated with early cognitive abilities (*β* = 0.51, *p <* 0.001) and was negatively affected by VP/VLBW and early motor difficulties (*β* = −0.12, *p =* 0.044 and *β* = −0.07, *p =* 0.010, respectively). The effects of VP/VLBW on adult IQ were partly mediated by early motor difficulties (*β* = −0.03, *p =* 0.023). Moreover, SES was associated with both child and adult IQ, but not with any of the other covariates (Table 2).

Measures of motor and cognitive function were associated in childhood (*r* = −0.27, *p < 0*.001) but not in adulthood (*r* = 0.02, *p =* 0.810). All predictors explained 13% of the variance in early motor difficulties and 14% in adult motor competence, 25% in child IQ, and 47% in adult IQ. Model fits suggest a fully saturated model and indicate a good statistical model fit once non-significant paths were removed (*χ^2^* (2) = 4.795, *p =* 0.091; CFI = 0.989; RMSEA = 0.058).

After excluding participants with neurological impairments (Model 2), the results of SEM model 2 (Table 3 and Figure 3) showed that early motor difficulties (*β* = 0.13, *p =* 0.029) were still associated with lower motor competence in adulthood but VP/VLBW was not (*β* = 0.10, *p =* 0.311), after controlling for IQ at 56 months, SGA, multiple births, child sex, and SES. VP/VLBW (*β* = 0.24, *p =* 0.001) and multiple births were associated with early motor difficulties (*β* = −0.16, *p =* 0.050).

Adult IQ was predicted by child IQ (*β* = 0.49, *p <* 0.001) and VP/VLBW (*β* = −0.13, *p =* 0.044), but was not affected by early motor difficulties (*β* = 0.02, *p =* 0.520). Moreover, the effects of VP/VLBW on adult IQ was not mediated by early motor difficulties (*β* = 0.00, *p =* 0.527). Family SES affected both child and adult IQ (Table 3).

Associations between motor and cognitive function were found in childhood (*r* = −0.23, *p <* 0.001) but not in adulthood (*r* = 0.09, *p =* 0.810). All predictors explained 9% of the variance in early motor difficulties and 8% in adult motor competence. Moreover, 24% and 42% of the variance in childhood and adult IQ were explained, respectively. After removing non-significant paths, the fit values of the fully saturated model indicated a good statistical model fit (*χ^2^* (3) = 3.493, *p =* 0.322; CFI = 0.997; RMSEA = 0.020).

## 5. Discussion

This prospective longitudinal study found that motor difficulties in early childhood were associated with later motor competence. Furthermore, early motor problems had a weak association with IQ in adulthood that was independent of the effects of VP/VLBW birth, and present after controlling for small for GA, multiple births, sex (male), family SES at birth, and preschool cognitive abilities (i.e., IQ at 56 months). However, once individuals with a neurological impairment at 56 months (which mostly included mild, stage 1 and 2 CP) were excluded, only the longitudinal associations between early motor difficulties and motor competence remained, but those with cognitive outcomes in adulthood were no longer significant.

The current findings show that children born VP/VLBW have a higher risk of motor difficulties during the preschool years, compared to term-born individuals. Once individuals with a neurological impairment were excluded, the risk was substantially reduced but VP/VLBW participants were still four times more likely to experience early motor difficulties. Similar prevalence rates for motor impairments in preterm children without CP have been reported previously [37,38].

Both VP/VLBW and term-born individuals with motor difficulties in infancy and toddlerhood (including those without neurological impairment) still had lower motor competence in adulthood [6,8,39]. Some previous studies have found that early motor development or aspects of early motor function are predictive of later motor skills in preterm [40,41], but not in healthy term-born children [42,43,44]. In contrast, other studies reported persistent motor problems from middle childhood to adulthood, in both preterm [6,8] and term populations [39,45]. In this study, persistent lower motor function was determined across four time points (from birth to 56 months), resulting in consistent long-term associations between early motor function and adult motor competence in both preterm- and term-born individuals. Rather than identifying individuals with persistent early motor problems, previous studies often took a snapshot, i.e., one single measure of motor functioning at a single time point.

In accordance with previous work, VP/VLBW birth and childhood IQ were associated with IQ in adulthood [7,46]. In addition to VP/VLBW birth and child IQ, early motor difficulties showed significant association with adult IQ. This may not be surprising as IQ, particularly performance IQ, is reliant on aspects of motor function or motor skills [47], and associations between motor functioning and cognitive outcomes have been demonstrated in previous studies of preterm children [48,49]. Importantly, the longitudinal associations between early motor difficulties and adult IQ in the current study were independent of childhood IQ and therefore cannot be solely attributed to an early link between motor and cognitive function (Figure 2). In adulthood, motor competence and IQ were not correlated, indicating that they measure two independent constructs. This may be explained by previously proposed concepts or theories of differentiation and integration, which posit that basic skills are integrated into new and higher-level functions. This can be manifested by a developmental path from motor to sensorimotor to cognitive function [50,51]. In adulthood, IQ assesses higher abstract logical reasoning and cognitive functioning that is not dependent on sensorimotor skills (as in early childhood). Considering the link between early motor difficulties and adult IQ found here (Figure 2), this may speak against this concept, and instead suggests global long-term associations between both domains, i.e., motor difficulties in early childhood predicting both motor competence and IQ in adulthood. However, once individuals who were diagnosed with a neurological impairment were excluded from analysis (Figure 3), early motor difficulties no longer predicted adult IQ. The most frequent neurological impairment was mild CP, which is often caused by bleeding in the brain such as periventricular leukomalacia that may affect wider areas of white matter rather than just the motor neurons and alteration of brain development [52]. Thus, both motor and cognitive abilities may share the same underlying cause. In contrast, in those without major neurological impairment, motor and cognitive function are related to different brain areas during development, i.e., differentiation, so that IQ in adulthood was not affected by early motor difficulties. As a result, the predictive models for both the motor and the cognitive domain were homotypic. Thus, early motor difficulties identified in the current study may serve as a predictor of later problems in motor competence in all children. In children who had been diagnosed with a neurological impairment, the prediction from early motor difficulties may be extended to short- and long-term problems in other developmental domains, such as academic achievement, mental health, and social functioning [21,53,54,55].

Overall, the findings of the current study suggest that motor competence in adulthood is independently, directly, and thus homotypically, predicted by early motor difficulties, while adult cognitive function is associated with early cognitive function and VP/VLBW birth. Motor and cognitive deficits have been identified as the most frequent adverse sequelae with the highest effect sizes differentiating VP/VLBW from term-born children and adults [9]. Therefore, it is important to test both motor and cognitive function as early as possible to identify children at risk for motor and cognitive sequelae.

The current study has several strengths. It is a prospective long-term follow-up study of a large whole population sample of VP/VLBW and term-born individuals recruited in the same hospitals using data based on age-appropriate physical and neurological examinations to assess early motor functioning, and reliable and valid tests to assess IQ in adulthood. There were some limitations to the current study. Although 47% of the eligible VP/VLBW and term-born individuals had complete data from birth to 26 years, the dropout was not random, as low SES families were less likely to continue participation [29]. As social factors are a major reason for dropout in most longitudinal studies, we controlled for family SES at birth in our analyses. However, to ascertain the validity of the results, it would be important to replicate the findings in a larger sample. Furthermore, whether the results of this study generalize beyond our at-risk sample requires testing.

## 6. Conclusions

Our results demonstrate that preterm-born individuals have a high risk of early motor difficulties. Early motor difficulties and cognitive problems uniquely predict later motor competence and IQ in adulthood, respectively. In children with neurological impairment, both motor and cognitive function are likely to be affected not only in childhood but also into adulthood. The results further demonstrate that while motor and cognitive functions are related in childhood, they differentiate into adulthood and are no longer related. The findings highlight the importance of testing motor and cognitive function in all children early on and to follow up the children that are identified to be at risk for adverse motor and cognitive outcomes.

## Figures and Tables

**Figure 1 ijerph-17-03258-f001:**
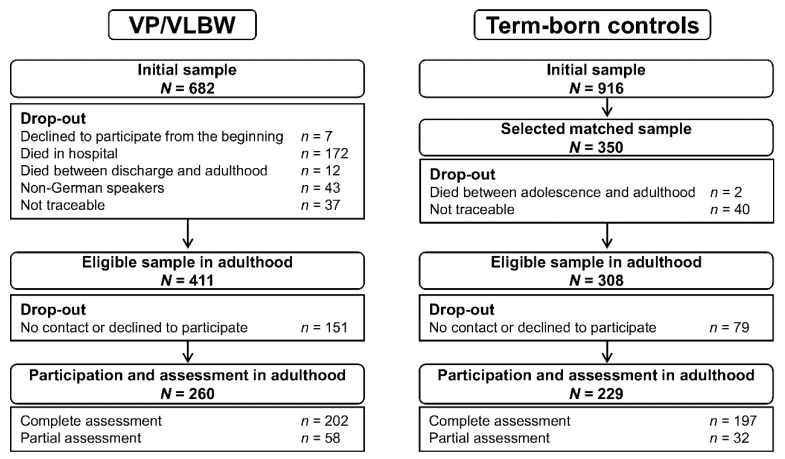
Eligible sample of very preterm and/or very low birth weight (VP/VLBW) and term control participants at age 26 years.

**Figure 2 ijerph-17-03258-f002:**
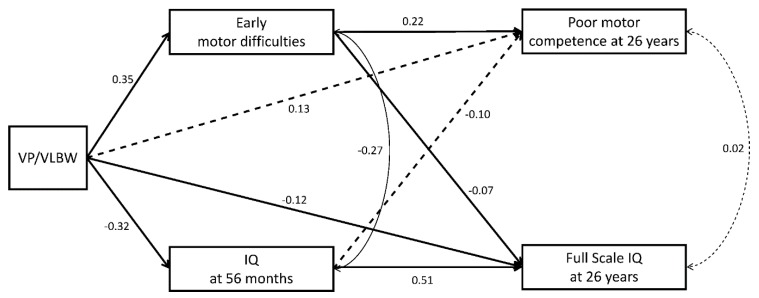
Structural equation model showing direct and indirect effects of VP/VLBW and early motor trajectories (difficulties) on motor competence and IQ at 26 years (Model 1). Bold lines represent hypothesized effects, solid lines represent significant effects, and dashed lines represent non-significant effects (standardized regression coefficients *β*). Effects of interest are shown. Effects of covariates and error terms are not presented to enhance readability. For effects of confounds see Table 2.

**Figure 3 ijerph-17-03258-f003:**
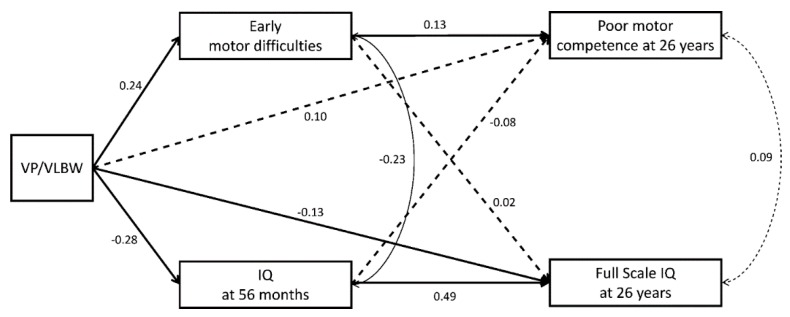
Structural equation model showing direct and indirect effects of VP/VLBW and early motor trajectories (difficulties) on motor competence and IQ at 26 years (excluding participants with neurological impairment, Model 2). Bold lines represent hypothesized effects, solid lines represent significant effects, and dashed lines represent non-significant effects (standardized regression coefficients *β*). Effects of interest are shown. Effects of covariates and error terms are not presented to enhance readability. For effects of confounds see Table 3.

**Table 1 ijerph-17-03258-t001:** Comparison of preterm and term control adults on neonatal characteristics, early childhood function, and motor competence and cognitive function in adulthood (*n* = 418).

Neonatal Characteristics, Early Childhood Function and Adult Outcomes	Term Controls	VP/VLBW	
	*n* = 226	*n* = 192	***p*-value**
Neonatal Characteristics			
Gestational Age (GA), in Weeks, Mean (SD)	39.66 (1.16)	30.51 (2.11)	**<0.001**
Birth Weight (BW), in Grams, Mean (SD)	3367 (441)	1344 (328)	**<0.001**
Small For GA <10%, *n* (%)	23 (10.2)	73 (38.0)	**<0.001**
Multiple, *n* (%)	6 (2.7)	50 (26.0)	**<0.001**
Sex, *n* (%)			0.059
Male	105 (46.5)	107 (55.7)	
Female	121 (53.5)	85 (43.3)	
Socioeconomic Status at Birth, *n* (%)			**0.010**
High	77 (34.1)	42 (21.9)	
Middle	95 (42.0)	85 (44.3)	
Low	54 (23.9)	65 (33.9)	
Early Childhood Function			
Early Motor Trajectories, *n* (%)			**<0.001**
Low Degree of Difficulties	222 (98.2)	152 (79.2)	
High Degree of Difficulties	4 (1.8)	40 (20.8)	
General Intelligence (IQ) at 56 Months, Mean (SD)	101.92 (12.94)	87.92 (19.82)	**<0.001**
Neurological Impairments at 56 Months, *n* (%)	2 (0.9)	23 (12.0)	**<0.001**
Epilepsy	1 (0.4)	2 (1.0)	0.470
Hydrocephalus	0 (0.0)	2 (1.0)	0.124
Cerebral Palsy (CP)	0 (0.0)	20 (10.4)	**<0.001**
CP Stage 1	0 (0.0)	10 (5.2)	
CP Stage 2	0 (0.0)	7 (3.6)	
CP Stage 3	0 (0.0)	3 (1.6)	
CP Stage 4	0 (0.0)	0 (0.0)	
Blindness	0 (0.0)	1 (0.5)	0.227
Deafness	1 (0.4)	0 (0.0)	0.356
Adulthood Outcomes at 26 Years			
Motor Competence, *n* (%)			**0.003**
High Motor Competence	204 (90.3)	154 (80.2)	
Low Motor Competence	22 (9.7)	38 (19.8)	
Cognitive Function, Mean (SD)			
General Intelligence (Full-Scale IQ)	102.62 (12.57)	91.21 (17.05)	**<0.001**

Note. Bold *p*-values were significant at the 0.05 level; Abbreviations: SD = standard deviation, VP/VLBW = very preterm and/or very low birth weight.

**Table 2 ijerph-17-03258-t002:** Regression coefficients using structural equation modelling (Model 1).

Direct and Indirect Effects	Unstandardized			Standardized		
	B	Standard Error	95% CI	β	*p*-Value	R^2^
Direct Effects						
Motor Competence at 26 Years						0.14
Motor Difficulties from Birth to 56 Months	0.77	0.22	(0.41, 1.13)	0.22	**<0.001**	
IQ at 56 Months	−0.11	0.07	(−0.23, 0.01)	−0.10	0.137	
VP/VLBW	0.28	0.20	(−0.05, 0.60)	0.13	0.151	
SGA	0.03	0.20	(−0.29, 0.35)	0.01	0.884	
Multiple Births	0.13	0.24	(−0.26, 0.53)	0.04	0.582	
Sex (Male)	−0.35	0.17	(−0.62, −0.07)	−0.16	**0.032**	
SES Low	−0.05	0.20	(−0.37, 0.27)	−0.02	0.794	
SES High	0.28	0.20	(−0.05, 0.60)	0.12	0.162	
Motor Difficulties from Birth to 56 Months						0.13
VP/VLBW	0.22	0.05	(0.14, 0.30)	0.35	**<0.001**	
SGA	0.03	0.03	(−0.02, 0.08)	0.05	0.291	
Multiple Births	−0.17	0.06	(−0.26, −0.08)	−0.19	**0.002**	
Sex (Male)	0.00	0.03	(−0.05, 0.05)	0.00	0.979	
SES Low	0.01	0.03	(−0.04, 0.07)	0.02	0.724	
SES High	−0.03	0.04	(−0.10, 0.04)	−0.04	0.482	
Full-Scale IQ at 26 Years						0.47
Motor Difficulties from Birth to 56 Months	−0.24	0.09	(−0.39, −0.08)	−0.07	**0.010**	
IQ at 56 Mon	0.51	0.04	(0.44, 0.58)	0.51	**<0.001**	
VP/VLBW	−0.25	0.12	(−0.45, −0.04)	−0.12	**0.044**	
SGA	−0.09	0.12	(−0.28, 0.10)	−0.04	0.424	
Multiple Births	0.11	0.13	(−0.10, 0.32)	0.04	0.384	
Sex (Male)	0.15	0.08	(0.01, 0.28)	0.07	0.075	
SES Low	−0.12	0.10	(−0.28, 0.04)	−0.05	0.220	
SES High	0.35	0.11	(0.17, 0.53)	0.16	**0.001**	
IQ at 56 Mon						0.25
VP/VLBW	−0.64	0.11	(−0.81, −0.46)	−0.32	**<0.001**	
SGA	−0.20	0.11	(−0.38, −0.03)	−0.09	0.059	
Multiple Births	−0.02	0.13	(−0.22, 0.19)	−0.00	0.922	
Sex (Male)	−0.01	0.09	(−0.16, 0.13)	−0.01	0.883	
SES Low	−0.45	0.11	(−0.63, −0.27)	−0.20	**<0.001**	
SES High	0.34	0.11	(0.17, 0.52)	0.16	**0.001**	
Indirect Effects						
Full-Scale IQ at 26 Years Via Motor Difficulties from Birth to 56 Months						
From VP/VLBW	−0.05	0.02	(−0.09, −0.01)	−0.03	**0.023**	

Note. Bold *p*-values were significant at the 0.05 level; Abbreviations: SGA = small for gestational age, SES = socioeconomic status.

**Table 3 ijerph-17-03258-t003:** Regression coefficients using structural equation modelling (excluding participants with neurological impairment, Model 2).

Direct and Indirect Effects	Unstandardized			Standardized		
	B	Standard Error	95% CI	Β	*p*-Value	R^2^
Direct Effects						
Motor Competence at 26 Years						0.08
Motor Difficulties from Birth to 56 Months	0.57	0.26	(0.13, 1.00)	0.13	**0.029**	
IQ at 56 Months	−0.09	0.08	(−0.21, 0.04)	−0.08	0.253	
VP/VLBW	0.21	0.21	(−0.13, 0.55)	0.10	0.311	
SGA	−0.07	0.22	(−0.43, 0.28)	−0.03	0.732	
Multiple Births	0.12	0.27	(−0.32, 0.56)	0.04	0.653	
Sex (Male)	−0.26	0.18	(−0.56, 0.04)	−0.13	0.142	
SES Low	0.12	0.21	(−0.23, 0.47)	0.05	0.566	
SES High	0.38	0.22	(0.02, 0.73)	0.16	0.076	
Motor Difficulties from Birth to 56 Months						0.09
VP/VLBW	0.12	0.04	(0.06, 0.18)	0.24	**0.001**	
SGA	0.05	0.03	(−0.00, 0.09)	0.08	0.103	
Multiple Births	−0.12	0.06	(−0.21, −0.02)	−0.16	**0.050**	
Sex (Male)	0.01	0.03	(−0.03, 0.06)	0.02	0.675	
SES Low	0.04	0.03	(−0.01, 0.08)	0.07	0.151	
SES High	−0.01	0.04	(−0.07, 0.05)	−0.01	0.850	
Full-Scale IQ at 26 Years						0.42
Motor Difficulties from Birth to 56 Months	0.07	0.11	(−0.10, 0.24)	0.02	0.520	
IQ at 56 Months	0.49	0.05	(0.41, 0.56)	0.49	**<0.001**	
VP/VLBW	−0.26	0.13	(−0.47, −0.05)	−0.13	**0.044**	
SGA	−0.10	0.13	(−0.32, 0.11)	−0.04	0.433	
Multiple Births	0.14	0.15	(−0.10, 0.38)	0.05	0.345	
Sex (Male)	0.12	0.09	(−0.03, 0.27)	0.06	0.171	
SES Low	−0.21	0.11	(−0.39, −0.02)	−0.09	0.069	
SES High	0.32	0.11	(0.14, 0.51)	0.15	**0.003**	
IQ at 56 Months						0.24
VP/VLBW	−0.57	0.11	(−0.75, −0.38)	−0.28	**<0.001**	
SGA	−0.16	0.11	(−0.35, 0.02)	−0.07	0.149	
Multiple Births	−0.12	0.13	(−0.33, 0.10)	−0.04	0.362	
Sex (Male)	−0.04	0.09	(−0.20, 0.11)	−0.02	0.664	
SES Low	−0.50	0.11	(−0.68, −0.31)	−0.22	**<0.001**	
SES High	0.33	0.11	(0.15, 0.52)	0.15	**0.002**	
Indirect Effects						
Full-Scale IQ at 26 Years Via Motor Difficulties from Birth to 56 Months						
From VP/VLBW	0.01	0.01	(−0.01, 0.03)	0.00	0.527	

Note. Bold *p*-values were significant at the 0.05 level.

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
