# Peer review of "Early Motor Trajectories Predict Motor but not Cognitive Function in Preterm- and Term-Born Adults without Pre-existing Neurological Conditions"

_ijerph, 2020, doi:10.3390/ijerph17093258_

Round 1

Reviewer 1 Report

The authors present an interesting finding in the long term follow up of pediatric subjects with early motor difficulties and their association with motor and cognitive outcomes in adulthood. The manuscript is well written.

The central theme of the paper is that early motor difficulties were associated with poorer motor competence in adulthood and weakly associated with adult IQ, but early motor difficulties ceased to be associated with adult IQ, once participants with neurological impairments were excluded (as demonstrated through the standardized regression coefficient changing from -0.07 to 0.02). 

My comments and questions to the authors are as below.

  1. The title is good, but the author should consider accommodating the confounder of preexisting "neurological problems" for their significant findings as this may be confusing to the readers as motor difficulties in early childhood are known to have a weak association with cognitive function in adulthood.
  2. The abstract is well written, but there is a minor typo in the conclusion of the abstract with repetition of the term "childhood" twice.
  3. Can the author describe and elaborate on the spectrum of early motor impairment/difficulties
  4. Most of the term children, which were controls probably manifest with no or low degree of motor difficulties across early childhood but categorizing them as a low or high degree of motor difficulties in term controls may give an incorrect impression to the readers.
  5. Were there any inclusion or exclusion criteria? How does the author exclude selection bias as there were few cerebral palsy patients, especially stage 2,3, and 4? Intraventricular hemorrhage is a common neurological problem affecting 15-20% of the babies born <32 weeks. 
  6. Can the authors provide comments on the near similar IQs with different testing at two different age spectrums (56 months and 26 years)?
  7. Table1, Figures 1 and 2 are nicely elucidating the significant results of the paper, and references are appropriate.
  8. Tables 2 and 3 should be modified for ease of reading or consider adding it as a supplemental table. 
  9. There is growing evidence to suggest that motor and cognitive development are interrelated, especially at an early age, but the authors present an interesting finding. Based on the explanation provided for no correlation between the cognitive outcome in adulthood and early motor difficulties when neurological impairments were excluded, it will be interesting to see the data of the mean IQ at 56 months and 26 years of subjects with and without neurological impairment.
  10. I applaud the author for this significant and exciting finding. Still, the author should emphasize the preliminary nature of the results as a multicenter study with a larger study cohort with and without neurological impairments is necessary to validate these results.

Reviewer 2 Report

The article by Baumann et al. analyses whether the motor development in childhood is associated with motor and cognitive outcomes in adulthood. The authors have conducted prospective Bavarian Longitudinal Study in 489 patients, which includes very preterm and term-born individuals to access early motor trajectories, adult motor, and cognitive outcomes. The authors observed that there is an increased risk for early motor difficulties in preterm or low birth weight individuals. Further, early motor difficulties were also associated with poorer motor competence in adulthood, but not with adult IQ. The authors also observed that early cognitive problems are associated with adult cognitive outcomes. These are the points that should be addressed by the authors.

  1. The authors should include the significance of this study in the conclusion section.
  2. The current study has taken into account the adulthood outcomes at 26 years of age. Is there any reason for choosing this particular age?
  3. Have authors considered any other factors which might contribute to the cognitive outcomes or motor competence in adults other than early motor difficulties?
  4. The authors should take into consideration if these adults have any pre-existing disease conditions and the effect of those conditions on their motor and cognitive skills.
  5. All the patients in this study are only restricted to Southern Bavaria that reduces the significance of the study. Do authors think that the results obtained by this study could be extrapolated worldwide?
  6. Authors have concluded that early cognitive and motor problems in childhood are associated with adult cognitive and motor outcomes respectively. According to the authors, how this issue could be addressed and what preventive measures could be taken. Explain in the discussion part.
  7. The language of the manuscript should be clearer and the writing should be improved. It is recommended to check the grammatical errors throughout the manuscript.

Reviewer 3 Report

The topic of the manuscript “Early motor trajectories predict motor but not cognitive function in adulthood” is a very interesting issue for the IJERPH readers. However, the authors should provide additional information which allows the reader to better understand the study interest and its results and, moreover, which allows the reproducibility by other researchers. We addressed additional comments to improve the quality of the paper.

  1. According to STROBE guidelines, the study design should be included in the title. 
  2. Abstract: In the "Introduction", the authors must minimally provide some context for the study performed, before mentioning the objective.
  3. Abstract: The “Results” epigraph should contain the quantitative results for the objectives that have been previously defined.
  4. Introduction: Information about the design type and assessment tools should not be included in this part (lines 57-61). These details should be transferred to the “Methods”
  5. Introduction: In this section, the authors must widely justify both the reasons and the scientific foundation behind the study.
  6. Methods: The manuscript should not include duplicate Information (lines 70-71 and 73-74).
  7. Methods: In the first epigraph, the authors must include a detailed description about the study design as well as about its key elements (framework, places, relevant dates, recruitment periods, tracing methods, data collection…).
  8. Methods: The information about the sample used in this work is not clear. The data provided is very confusing to the reader. Authors should consider using a flow chart to capture information about participants in each phase of the study: baseline cohort, eligible recruited participants, etc. (lines 71-78).
  9. Methods: Please, include more information about the guidelines followed and the number of the register of the ethic commit.
  10. Methods: In the “Measures” section, specifically right after the “Early motor trajectories” subsection, a new epigraph which describes the measures of cognitive function in childhood must be included. This should be replaced by the content of lines 121-123 in the “Outcomes in adulthood” subsection. It should also include an explanation about the assessment instruments used.
  11. Methods: Was the time of stay in the VP / VLBW NICU not collected as a covariable? A justification about the reason why this variable was not usted must be included.
  12. Methods: The data collection about “Neurological impairments in early childhood” (Point 3.4) should be included in point 3.3 as covariables. Likewise, the authors must define the units or categories in which the covariables are collected.
  13. Statistical Analyses: It is not clarified if the authors have verified the data normality in order to know if the data should be shown as means or medians and parametric or non-parametric statistics.
  14. Statistical Analyses: The authors should include the reasons for including the confusion factors for which they are adjusted (line 139-140).
  15. Statistical Analyses: The duplicated Information provided must be reviewed (lines 139-140 and 146-148).
  16. Statistical Analyses: The authors should clarify if they have performed sensitivity analyses (line 149).
  17. Results: The authors should expand the information that describes the main characteristics of the study participants and support it through clarifying quantitative information, such as prevalences and frequencies. In Table 1, the authors must indicate next to each variable how the data is expressed (n (%) or mean (SD)). Furthermore, an explanation about the diagnosis stages of cerebral palsy must be included in the corresponding "Measures" section. Finally, the extensive explanation in the footer shall be removed.
  18. Discussion: In the limitation of the study should be include information about the select bias or other residual confounding or nor causal effects which can affect to internal validity. Furthermore, the generalizability of the study results (external validity) should be discussed.
  19. The authors should include a brief section summarizing the implications of work for practice and research.

Round 2

Reviewer 2 Report

The authors have satisfactorily addressed all the concerns.